# Fabrication and Characterization of Cellulose Nanofiber Aerogels Prepared via Two Different Drying Techniques

**DOI:** 10.3390/polym12112583

**Published:** 2020-11-03

**Authors:** Zhe Wang, Wenkai Zhu, Runzhou Huang, Yang Zhang, Chong Jia, Hua Zhao, Wei Chen, Yuanyuan Xue

**Affiliations:** Co-Innovation Center of Efficient Processing and Utilization of Forest Products, College of Materials Science and Engineering, Nanjing Forestry University, Nanjing 210037, China; wangzhe_njfu@163.com (Z.W.); wenkai1992@njfu.edu.cn (W.Z.); yangzhang31@126.com (Y.Z.); cctv_jc_2000@126.com (C.J.); forestry2020@163.com (H.Z.); c1246700079@163.com (W.C.); xuzhenyuan_1993@163.com (Y.X.)

**Keywords:** cellulose nanofiber (CNF) aerogel, supercritical CO_2_ drying, liquid nitrogen freeze-drying, performance characteristics

## Abstract

Studies on the influence of drying processes on cellulose nanofiber (CNF) aerogel performance has always been a great challenge. In this study, CNF aerogels were prepared via two different drying techniques. The CNF solution was prepared via existing chemical methods, and the resultant aerogel was fabricated through supercritical CO_2_ drying and liquid nitrogen freeze-drying techniques. The microstructure, shrinkage, specific surface area, pore volume, density, compression strength, and isothermal desorption curves of CNF aerogel were characterized. The aerogel obtained from the liquid nitrogen freeze-drying method showed a relatively higher shrinkage, higher compression strength, lower specific surface area, higher pore volume, and higher density. The N_2_ adsorption capacity and pore diameter of the aerogel obtained via the liquid nitrogen freeze-drying method were lower than the aerogel that underwent supercritical CO_2_ drying. However, the structures of CNF aerogels obtained from these two drying methods were extremely similar.

## 1. Introduction

Aerogel is a porous material obtained by replacing the liquid component of a gel with gas without changing the three-dimensional network structure or volume of the gel [1,2]. Besides, aerogels offer several advantages, including high porosity, large specific surface area, low density, low thermal conductivity, low dielectric constant, and a unique three-dimensional network structure [3,4,5]. Therefore, aerogels can be potentially applied in civil, military, aerospace, and other high-tech fields [6]. Nanocellulose-based aerogels are known for their renewable, abundant reserves and cost efficiency [7,8,9]. Most importantly, cellulose aerogels, as the third generation of new materials, possess the characteristics of traditional aerogels but surpass the features of inorganic and organic polymer aerogels. Due to their advantages of regeneration, degradation, and biocompatibility, cellulose aerogels find a broad range of applications [10,11,12,13].

A large number of exposed hydroxyl groups on the surface of nanocellulose-based wet gel promotes the formation of inter- and intramolecular hydrogen bonds [14,15]. Additionally, nanocellulose can be gelated to form a three-dimensional network structure of physical cross-linking [16,17]. Generally, nanocellulose aerogel materials can be obtained by replacement of liquid solution in wet gel by air via supercritical CO_2_ drying, freeze-drying, and other technologies [18,19,20]. Therefore, the drying method significantly influences the formation of nanocellulose aerogels. The drying methods directly determine the formation and quality of aerogels. As one of the most widely used nanocellulose aerogels, the drying methods of cellulose nanofiber (CNF) aerogels have been extensively studied [21,22,23,24]. Wang et al. [25] prepared nanocellulose aerogels via supercritical CO_2_ drying following the addition of calcium chloride solution to the nanocrystalline cellulose colloid. The resultant aerogels exhibited a highly porous network with high specific surface area and low shrinkage rate. Based on the formation of nanofibrous structure in biopolymer aerogels through supercritical CO_2_ processing, Takeshita et al. [26] investigated the formation of a nanofibrous microstructure in chitosan aerogel. It was proven that postgelation treatment of hydrogels, particularly solvent exchange with CO_2_ at the initial stage of the supercritical drying process, is the key step in the formation of the nanofibrous aerogel structure. Baldino et al. [27] produced bicomponent natural aerogels alginate-gelatin (A/G) and chitosan-gelatin (CS/G) by supercritical gel drying. Supercritical gel drying preserved their delicate nanostructured morphology, and all the aerogels were characterized by porosity values higher than 80%. Wan et al. [28] successfully obtained a graphene-cellulose mixed aerogel via the freeze-drying of regenerated mixed solvent. The freeze-dried hybrid aerogels shrunk and exhibited excellent dimensional stability and complete pore structure. Besides, they showed excellent electromagnetic interference resistance and electrical conductivity. Therefore, this aerogel material finds potential application in the electrical industry. The commonly used drying methods include supercritical CO_2_ drying and freeze-drying techniques. The supercritical CO_2_ drying method can be done at room temperature. The resulting aerogel has a low shrinkage. The skeleton structure has small damage and uniform pore size distribution. The disadvantages are high cost, high equipment requirements, and long experiment cycle. Cellulose aerogels obtained by freeze-drying have good quality and low shrinkage. The disadvantages are high equipment and process requirements as well as the fact that the solvent cannot be chosen at will. Previous studies have shown that the incorporation of different types of cellulose nanofibers produce significant differences in the formation of the corresponding aerogels. However, the effect of different drying processes on the same cellulose nanofibers has not been studied [29,30]. Therefore, different drying methods are used in this study to obtain CNF aerogels.

In this study, the supercritical CO_2_ drying and liquid nitrogen freeze-drying techniques were incorporated to prepare spherical CNF aerogels. The physical and chemical properties of the aerogel materials were studied (e.g., shrinkage, specific surface area, isothermal desorption curves, and microstructure) to compare the characteristics of the two drying methods. This study is expected to provide an effective reference for the preparation of high-performance cellulose nanofiber aerogel products.

## 2. Experimental

### 2.1. Materials

Needle wood pulp fiber (after the removal of hemicellulose and lignin) was purchased from a paper mill (Jiangsu, China). Ammonium sulfate ((NH_4_)_2_SO_4_, ≥99.5% purity) and calcium chloride (CaCl_2_, 95% purity) were obtained from Anhui Sunhere Pharmaceutical Excipients Co., Ltd. (Hefei, China). The dialysis membranes (3 inches wide, MWCO: 8000–14,000 Da) were purchased from Nanjing Chemical Reagent Co., Ltd. (Nanjing, China). Distilled water was used for the preparation of solutions in all the experiments.

### 2.2. Preparation of CNF

A schematic illustration of the preparation of CNF aerogels via the two drying methods is shown in Figure 1. CNF was obtained by oxidizing the pulp with ammonium persulfate. Briefly, 3 g of needle wood pulp and 300 mL of ammonium persulfate with a concentration of 1.5 mol·L^−1^ prepared in the laboratory were taken in a flask. The mixture was stirred magnetically (300 rpm/min) for 16 h under 70 °C and diluted with distilled water (500 mL). Later, the reaction was terminated by stirring the mixture with a glass rod for 5 min.

### 2.3. Preparation of CNF Hydrogels

In this study, CaCl_2_ (inorganic salt) solution was used as an inorganic salt-assisted gel to prepare CNF hydrogels. First, 0.25 mol/L of CaCl_2_ solution was prepared in the laboratory. CNF suspensions with mass fractions of 1.5%, 2.5%, and 3.5% were prepared by weighing out 10 g of CNF. The prepared CNF suspensions were treated with an ultrasonic wave breaker in an ice bath for 15 min and further kept at room temperature for half an hour. Each of the CNF suspensions was slowly dropped into a Petri dish containing CaCl_2_ solution using a glass syringe. The Petri dishes were kept undisturbed for 48 h until gelation was complete. The dripping CNF suspension formed spherical hydrogels under the influence of CaCl_2_ solution.

### 2.4. Preparation of CNF Aerogels by Supercritical CO_2_ Drying

Supercritical CO_2_ drying of the CNF hydrogels was performed in an extraction autoclave under a pressure of 120 bar. CNF hydrogels of different concentrations (1.5%, 2.5%, and 3.5%) were placed in the extraction autoclave containing pure CO_2_ under a temperature of 40 °C (SFT-105, American SFT Company, Newark, DE, USA). The obtained CNF aerogels were labeled as 1.5-SCD-CNF aerogel, 2.5-SCD-CNF aerogel, and 3.5-SCD-CNF aerogel.

### 2.5. Preparation of CNF Aerogels by Liquid Nitrogen Freeze-Drying

The liquid nitrogen freeze-drying technique was performed in a freeze-dryer (FD526, American GOLD-SIM Company, Miami, FL, USA). Different concentrations (1.5%, 2.5%, and 3.5%) of CNF hydrogels were placed in copper tubes. The hydrogels were quickly frozen in liquid nitrogen. Subsequently, the frozen CNF hydrogels were dried in a freeze-dryer for 48 h to obtain CNF aerogels. The products were labeled as 1.5-FDN-CNF aerogel, 2.5-FDN-CNF aerogel, and 3.5-FDN-CNF aerogel.

### 2.6. Characterization

The chemical structures of the pulp and CNF were obtained using Fourier transform infrared spectroscopy (FTIR, VERTEX 80V, Brucker, Billerica, MA, USA). Samples were mixed with potassium bromide (KBr) in a mass fraction of 1:10. Infrared spectrum of the samples was measured in the range of 4000–500 cm^−1^ at a resolution of 6 cm^−1^. Scanning electron microscopy (SEM, JSM-7600, JEOL, Akishima, Japan) was used to analyze the surface morphology of the CNF aerogels. Before the test, thin layers of gold were sprayed on the CNF aerogels by sputter deposition at a low deposition rate. The test was performed in a vacuum under a working voltage of 20 kV. Transmission electron microscopy (TEM, JEM-1400, JEOL, Akishima, Japan) was used to capture structural images of the CNF. For TEM analysis, the CNF hydrogel was treated with distilled water and dispersed evenly after ultrasonic treatment. The samples were dropped onto a glow-discharged carbon-coated copper grid. TEM analysis was performed at an acceleration voltage of 200 kV. At least 15 TEM images of the CNF were randomly selected and analyzed using Image-Pro Plus 6.0 (Media Cybernetics, Inc., Rockville, MD, USA) software to determine the size distribution of the CNF. X-ray diffraction analyses of the samples were performed using Philips X ‘Pert Pro MPD X-ray diffractometer from the Netherlands. The analysis was performed by placing the film in a sample rack of glass under stable conditions. The test conditions were as follows: Cu Kα rays, Ni chip filter, λ = 0.154 nm; scanning range: 2q = 8–40°; and step scan: 0.1°/s. The synergies and drying shrinkage of the CNF aerogels were calculated using the following formula:(1)$$S=\frac{\left(L_{0}-L_{C}\right)}{L_{0}}\times 100\%$$
where *S* is the shrinkage rate of the aerogel, *L*_0_ represents the maximum average size of the hydrogel (mm), and *L*_c_ is the maximum average size of the aerogel (mm). Because the CNF aerogel did not form a perfect sphere, the change in the maximum diameter of the sample was measured. At least 50 balls were measured to calculate the average size of the aerogel. The specific surface area and pore volume of different concentrations of CNF aerogels prepared under the optimal drying process were determined using the automatic specific surface area analyzer (ASAP2020, Micrometrics, Norcross, GA, USA). Prior to the test, small pieces of the CNF aerogel samples were placed in the sample tube and degassed under vacuum at 90 °C for 8 h. Nitrogen was injected throughout the test under a temperature of 77 K. The apparent volumetric mass density of the CNF aerogels were determined by the drainage method and calculated by dividing their weight by volume, as shown in the following formula:(2)ρ=mV

The compression performance of different concentrations of CNF aerogel samples was evaluated using the universal mechanical testing machine model (CMT-4204, Shanghai, China). During the test, the compression speed was 1 mm/min, which was stopped as the sample was compressed to 70% of the initial diameter. The isotherms were obtained at 77 K using the Micrometrics ASAP 2020 analyzer (Micrometrics, MAC Inc., Norcross, GA, USA). The Brunauer-Emmett-Teller (BET) equation was used to calculate the specific surface area of the aerogels. The Barrett-Joyner-Halenda (BJH) method was used to calculate the mean aperture of the isothermal desorption branch. Before the test, the CNF aerogels were held in a vacuum overnight under room temperature. The pore size distribution of the CNF aerogels was obtained from nitrogen adsorption/desorption isotherms using the BJH equation.

## 3. Results and Discussion

### 3.1. FT-IR, TEM, and XRD of CNF

FT-IR spectroscopy was used to characterize the chemical bonds of the pulp and CNF. The infrared spectra of the raw material (pulp) and CNF obtained in this study are shown in Figure 2. As can be seen, the FT-IR spectrum indicated that the prepared CNF had the same characteristic cellulose absorption peak as the pulp. The characteristic absorption peaks at 3405, 2900, 1430, 1162, 1110, and 890 cm^−1^ seen in Figure 2a can be attributed to the cellulose I_β_ structure [31]. The peaks at 1635 cm^−1^ represent the bending vibration peak of –OH in water molecules. This was mainly caused by the absorption of water molecules by the hydrophilic groups in the cellulose molecules [32]. In addition, the peaks around 1202, 1162, and 1060 cm^−1^ represent the shear vibration peak, asymmetric vibration peak, and stretching vibration peak generated by CH_2_, C–O–C, and C–O on the cellulose epoxy structure C_6_, respectively [33,34]. The sharp peaks observed at 1370 cm^−1^ represent the C–H bending vibration. The peak at 1721 cm^−1^ can be attributed to the C=O stretching vibration peak of the carboxyl group. The –O–O– bond in ammonium persulfate is very unstable and can be broken down to form hydrogen peroxide and oxygen atoms with a highly oxidizing nature under heating conditions. This could have oxidized the primary alcohol hydroxyl group in the cellulose molecules into carboxyl groups, such as aldehyde.

The particle size distribution diagram and microscopic morphology image of the CNF are shown in Figure 3a,b, respectively. It can be seen from Figure 3a that the diameter of the CNF was mainly distributed between 50 and 70 nm, reaching the nanometer range in width. Furthermore, it can be seen from Figure 3b that the morphology of the CNF was mainly in the form of long fibers with a length of 1–2 µm and a length/diameter ratio of 15:40. The X-ray diffractograms of the pulp fiber and CNF were obtained as shown in Figure 3c. The CNFs showed the main characteristic peak angle of 23.1° (002) and two smaller peaks of 15.8° and 35.1°, respectively. The diffractograms displayed a mixture of polymorphs of cellulose I (typical peaks at 2θ of 15° and 22°) and cellulose II. Cellulose I and cellulose II, named as native and regenerated cellulose, respectively, are the most common polymorphs of the cellulose of interest. On comparing the intensities of the diffraction peaks of the two samples, it was possible to determine a change in the crystallinity. The intensity of the peaks in the diffractogram of CNF was significantly higher than those of the pulp. Thus, acid hydrolysis treatment increased the crystallinity. The results indicate that the amorphous regions were removed by acid hydrolysis.

### 3.2. Morphological Characterization

Figure 4 shows the microscopic morphology of CNF aerogels resulting from the two drying methods that were analyzed by the SEM technique. CNF aerogels of different mass fractions were obtained by the supercritical CO_2_ drying technique. The internal micromorphologies of the aerogels were observed using a scanning electron microscope. It can be seen from Figure 4a–c that the SEM images of the CNF aerogel showed a three-dimensional network structure without the collapse phenomenon. In the gel process, the CNF, being the basic framework, was surrounded by hydrogen bonds resulting from the hydroxyl groups to form a three-dimensional network structure [35]. The disappearance of the gas–liquid interface and surface tension caused by the supercritical CO_2_ drying process avoided destruction of the internal structure of the hydrogel. This indicated the considerable applicability of the supercritical CO_2_ drying process. For the CNF mass fraction of 1.5%, the pore structure inside the aerogel was obviously sparse, as seen from Figure 4a–c. For the mass fractions of 2.5% and 3.5%, the internal structure was relatively dense, even though there was no significant difference. Therefore, the concentration of CNF only affected the density and pore size of the inner network structure of the aerogel without affecting its morphology.

The internal micromorphology of the three CNF aerogels with different mass fractions achieved via the liquid nitrogen freeze-drying process is shown in Figure 4d–f. The internal structures of the spherical CNF aerogels obtained by liquid nitrogen freeze-drying were extremely similar. The CNF formed a three-dimensional network structure through hydrogen bond aggregation in the gelation process. During the freezing of liquid nitrogen, the CNF hydrogels were changed rapidly from liquid to crystal form. Figure 4d–f shows that the CNF aerogel had a three-dimensional porous layered structure composed of mesoporous resulting from the destruction of the three-dimensional network structure in the liquid nitrogen freeze-drying process. The liquid present in the pore structure was gradually sublimated and replaced by air in the liquid nitrogen freeze-drying process. The whole process was performed in a vacuum under low temperature (25 Pa, −55 °C). The surface tension generated by CNF hydrogel sublimation was minimal, which effectively preserved the layered three-dimensional porous structure of the CNF aerogel and slowed down sample shrinkage. Besides, there were tubular structures inside the CNF hydrogel. The analysis showed the formation of columnar crystals in the hydrogel during rapid freezing of the solvent. In addition, the shape of the crystals defined the template of the porous structure. As the solvent sublimated, the cylindrical ice crystals left behind these tubular structures. It can be clearly observed from the SEM images that the three-dimensional network structure remained on the sheet structure. Figure 4f shows that a small amount of three-dimensional network structure was not destroyed during the crystallization process. The sublimation of the hydrogels caused the formation of low surface tension, thus preserving the three-dimensional network structure to a small extent.

### 3.3. Analysis of Shrinkage, BET, Pore Volume, and Density

Lightweight CNF aerogels were obtained via supercritical CO_2_ drying and liquid nitrogen freeze-drying techniques. Table 1 shows the properties of different concentrations of CNF aerogels fabricated by the two drying techniques. It can be seen from Table 1 that the shrinkage rates of the CNF aerogels dried by the supercritical CO_2_ drying method were 5.56%, 4.12%, and 4.35%. The hydrogels with a mass fraction of 1.5% showed higher shrinkage than those with mass fractions of 2.5% and 3.5%. In addition, there was no significant difference in shrinkage for the hydrogels with mass fractions of 2.5% and 3.5%. Therefore, the analysis indicated that the mass fraction was related to the pore structure and strength of hydrogels. Moreover, the specific surface area and density of the CNF aerogels were increased with the increase in the mass fraction of CNF. The pore volume decreased with the increase in the mass fraction. Based on the BET specific surface area, the values of the specific surface area of the hydrogels were measured. The specific surface area values of the aerogels were determined from the three-dimensional network structure. In the gelation process, the cellulose forms intermolecular hydrogen bonds resulting from the hydroxyl groups as it interweaves to generate a three-dimensional network structure [36]. With the increase in the concentration of CNF, there was an enhancement in the number of hydroxyl groups in the cellulose, resulting in the formation of a more compact three-dimensional network structure. Therefore, the specific surface area of the CNF aerogels increased and the pore volume decreased with an increase in the mass fraction of the CNF. However, the specific surface area did not always increase with concentration. The extent of increase was eventually reduced as the specific surface area approached the maximum value. Moreover, the specific surface area decreased with concentration. When the mass fraction of CNF was greater than 3.5%, the spherical hydrogels were not formed. Therefore, there was no formation of hydrogels with an increase in the mass fraction values. The specific changing trends need to be further studied. The density of the aerogels increased with the mass fraction of CNF. The research reveals that the mass of CNF per unit volume increased with the increase of concentration, and the density thus increased.

It can be seen from Table 2 that the condensation rate of the CNF aerogels by the liquid nitrogen freeze-drying process decreased with the expansion of the mass fraction of CNF. When the mass fraction of CNF was 1.5%, the drying shrinkage of the hydrogel was found to be maximum. When the mass fractions of CNF were 2.5% and 3.5%, the drying shrinkage of the hydrogels was similar. A similar variation in the shrinkage rate was observed for hydrogels resulting from the supercritical CO_2_ drying process. For the same mass fraction of CNF, the shrinkage rate of the liquid nitrogen freeze-drying technique was higher than that of the supercritical CO_2_ drying process. This was due to the difference in the mechanism of the two freeze-drying methods. In addition, the specific surface area increased with the mass fraction of CNF, but the rate of elevation gradually decreased. Moreover, the pore volume decreased, and the density increased with the expansion of the mass fraction of CNF. In the freeze-drying process, the inner network structure of the CNF aerogel was denser with the extension of the CNF mass fraction, which further enhanced its strength. Therefore, the specific surface area and density increased as the pore volume decreased. For the same mass fraction of CNF, the specific surface area, pore volume, and density of the aerogels obtained by the liquid nitrogen freeze-drying process were reduced compared to the supercritical CO_2_ drying method. This can be mainly attributed to the high shrinkage of CNF aerogel under the liquid nitrogen freeze-drying technique, which destroyed the three-dimensional network structure within the aerogel.

### 3.4. Compression Strength

It can be seen from Figure 5 that the mechanical properties of the CNF aerogel enhanced with the increase in the mass fraction of CNF. The compression strength values of the aerogels with the mass fractions of 1.5%, 2.5%, and 3.5% prepared by the supercritical CO_2_ drying process were found to be 0.37, 0.67, and 0.77 MPa, respectively, when the compression ratio was 70%. In the liquid nitrogen freeze-drying process, the aerogels with the mass fractions of 1.5%, 2.5%, and 3.5% showed compression strength values of 0.85, 1.32, and 1.47 MPa, respectively, at a compression ratio of 70%. This can be mainly attributed to the presence of a large number of hydroxyl groups on the CNF surface, resulting in the formation of hydrogen bonds in the gelation process and cross-linking within the hydrogels to form a three-dimensional network structure [37]. Within a certain range, the number of hydrogen bonds formed between the cellulose in the hydrogels increased with concentration. The enhancement of physical cross-linking between each other resulted in the formation of a more compact internal network skeleton. Therefore, the mechanical properties of the aerogels enhanced with the increase in concentration. As the mass fraction of CNF ranged from 1.5% to 2.5%, there was greater enhancement in the compression strength of the aerogel compared to those with the CNF mass fractions in the range of 2.5% to 3.5%. The analysis showed that the internal structure of the aerogel was relatively loose for low mass fraction values of CNF. The increase in concentration resulted in a more compact internal structure. However, there was no obvious improvement in the internal structure as the density reached the maximum after a certain value of the mass fraction. For the same mass fraction, the aerogel compression strength after liquid nitrogen freeze-drying was greater than that under the supercritical CO_2_ drying process. This may be attributed to the formation of ice crystals by the CNF hydrogels during the freeze-drying method that changed the internal pore structure of the aerogel. Due to cellulose aggregation, the compression strength was increased.

### 3.5. Analysis of N_2_ Adsorption/Desorption Isotherms

N_2_ adsorption/desorption isotherms of the CNF aerogel after performing the two drying methods under optimal process conditions are shown in Figure 6. Figure 6a shows the relationship between the N_2_ adsorption/desorption isotherms of the CNF aerogel obtained by the supercritical CO_2_ drying method. Figure 6b shows the relationship between the adsorption isotherms of the CNF aerogel fabricated by the liquid nitrogen freeze-drying technique. It can be seen from Figure 6 that the isothermal stripping curves of the two drying methods were consistent with the type II adsorption isotherms [38]. Figure 6a shows that the adsorption capacity obviously increased with the increase in relative pressure in the low-pressure region (0–0.1). This indicated that the gas interacted strongly with the CNF aerogel surface, resulting in monolayer adsorption. Microporous structures were present in the CNF aerogel. The adsorption capacity was slowly increased with relative pressure. In addition, single-layer adsorption turned into multilayer adsorption in the range of moderate relative pressure values (0.2–0.8). The adsorption capacity augmented sharply at the high relative pressure region (0.8–1). The whole process was dominated by multilayer adsorption. It can be seen from Figure 6b that the adsorption capacity increased extremely slowly under low relative pressures (0–0.1), indicating the formation of a small number of micropores in the CNF aerogel. Subsequently, the adsorption capacity increased sharply with an increase in phase pressure. The adsorption hysteresis loop also belonged to the type II adsorption isotherm. This indicated that the mesopores inside the CNF aerogel were primarily dried by the two methods. It was also proven that the CNF aerogel was composed of a three-dimensional network structure formed by the self-aggregation of hydroxyl groups on the CNF crystal via hydrogen bonds.

The N_2_ adsorption capacity of the aerogel obtained by liquid nitrogen freeze-drying was lower than that resulting from the supercritical CO_2_ drying technique without any microporous structure. Additionally, the adsorption capacity of the CNF aerogels by the liquid nitrogen freeze-drying method increased gradually with the addition of CNF mass fraction. The agglomeration effect of CNF after the liquid nitrogen freeze-drying process led to the formation of a layered structure. Therefore, the porosity of the CNF aerogel was increased with the elevation in mass fraction. However, there was a relatively saturated mass fraction. The volume occupied by CNF itself affected the porosity of the material. Therefore, the adsorption capacity of the aerogel showed a gradual and stable trend with the increase in CNF concentration.

### 3.6. Analysis of the Pore Size Distributions

Based on the N_2_ adsorption/desorption isotherms of the CNF aerogels, the pore size distributions of the aerogel can be calculated using the BJH equation. Figure 7a shows the pore diameter distributions of the CNF aerogel by the supercritical CO_2_ drying technique containing different CNF mass fractions. Figure 7b shows the pore size distributions of different mass fractions of CNF aerogels prepared via the liquid nitrogen freeze-drying method. The pore diameter of the CNF aerogels dried by the supercritical CO_2_ drying method was mainly distributed at 33 and 23 nm, as seen from Figure 7a. Moreover, Figure 7b shows that the pore size distributions of the CNF aerogels by liquid nitrogen freeze-drying was mainly distributed at 23 and 18 nm. The pore diameter of the CNF aerogel was mainly mesoporous (2–50 nm) with few micropores (less than 2 nm). The CNF aerogels with the mass fractions of 2.5% and 3.5% showed similar pore size distributions. However, these pore sizes were lower than that of the aerosols showing the mass fraction of 1.5%. This can be attributed to the loose pore structure of the CNF aerogel with the mass fraction of 1.5% exhibiting a relatively large pore diameter. Generally, when the mass fractions were 2.5% and 3.5%, the pore structure inside the aerogels was found to be highly dense. The increase in mass fraction showed no remarkable influence on the pore diameter. The diameter of the aerogel obtained by the liquid nitrogen freeze-drying method was smaller than that obtained by the supercritical CO_2_ drying process possessing the same mass fraction. This may be attributed to the different drying principles involved in the two methods. Additionally, the liquid nitrogen freeze-drying method showed a high shrinkage rate.

## 4. Conclusions

In this study, CNF aerogels were prepared via the supercritical CO_2_ drying and liquid nitrogen freeze-drying techniques. Compared to supercritical drying, the CNF aerogels obtained from liquid nitrogen freeze-drying showed excellent shrinkage, lower specific surface area, higher pore volume, and higher density. The compression strength of the aerogel after the freeze-drying process was higher than that after the supercritical drying method. The N_2_ adsorption capacity of the aerogel obtained from freeze-drying was lower than that derived from supercritical CO_2_. The pore diameter of the aerogel prepared by the freeze-drying method was smaller than that obtained by the supercritical drying process. Furthermore, the internal structure of the spherical CNF aerogels obtained from the liquid nitrogen freeze-drying process was extremely similar. This work provides a novel idea for the preparation of high-performance CNF aerogels that are significant in the study of adsorption.

## Figures and Tables

**Figure 1 polymers-12-02583-f001:**
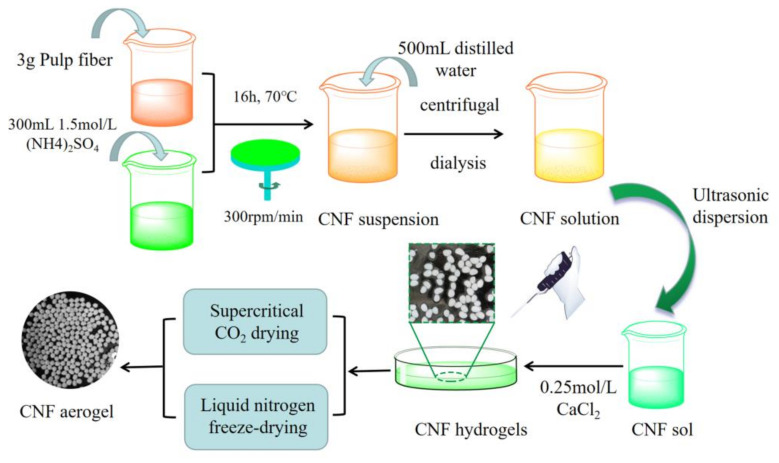
Schematic illustration of the preparation processes for cellulose nanofiber (CNF) aerogel via two drying methods.

**Figure 2 polymers-12-02583-f002:**
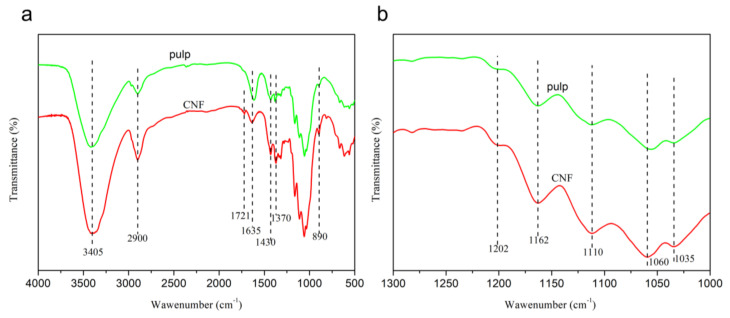
FTIR spectra of the pulp and CNF in the wave number range of 4000–500 cm^−1^ (**a**) and 1300–1000 cm^−1^ (**b**).

**Figure 3 polymers-12-02583-f003:**
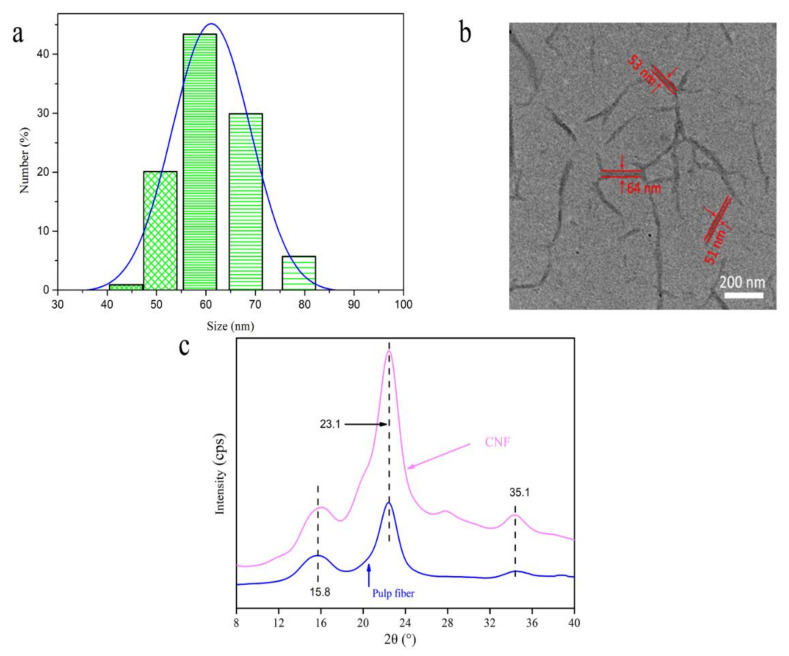
The size distribution (**a**), TEM image (**b**), and XRD spectra (**c**) of CNF.

**Figure 4 polymers-12-02583-f004:**
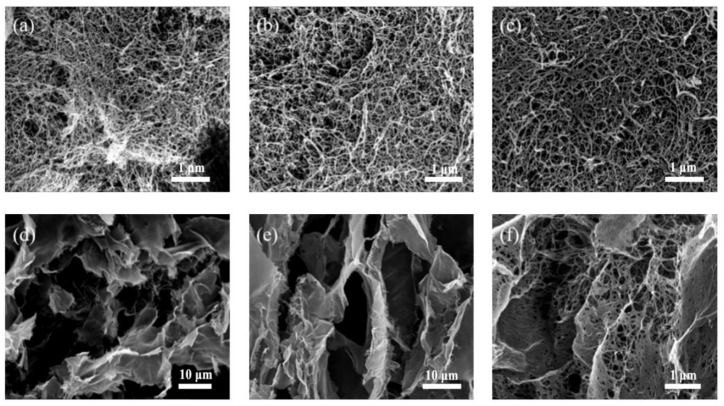
Microcosmic images of CNF aerogels: (**a**) 1.5-SCD-CNF aerogel, (**b**) 2.5-SCD-CNF aerogel, (**c**) 3.5-SCD-CNF aerogel, (**d**) 1.5-FDN-CNF aerogel, (**e**) 2.5-FDN-CNF aerogel, and (**f**) 3.5-FDN-CNF aerogel.

**Figure 5 polymers-12-02583-f005:**
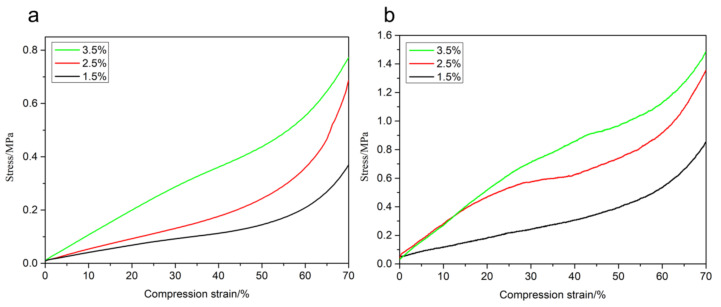
Diagram of the relationship between compressive strength and CNF concentration in the aerogel via two kinds of drying methods: (**a**) supercritical CO_2_ drying method and (**b**) liquid nitrogen freeze-drying technology.

**Figure 6 polymers-12-02583-f006:**
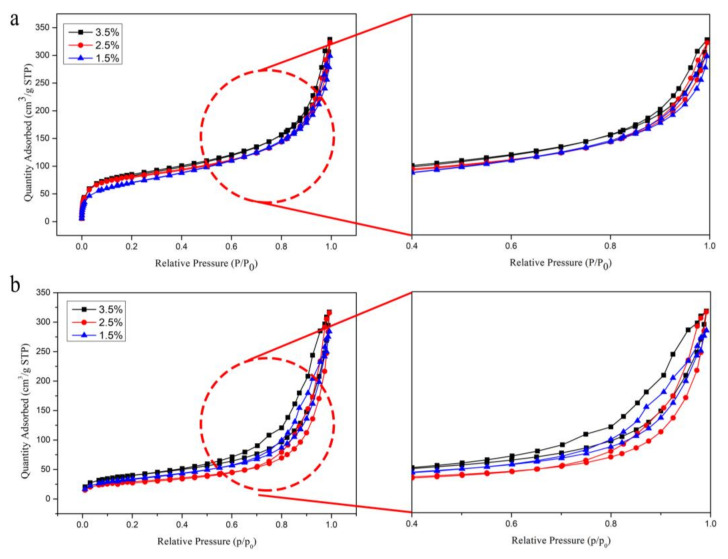
The N_2_ adsorption/desorption isotherms of CNF aerogel via two kinds of drying methods: (**a**) supercritical CO_2_ drying method and (**b**) liquid nitrogen freeze-drying technology.

**Figure 7 polymers-12-02583-f007:**
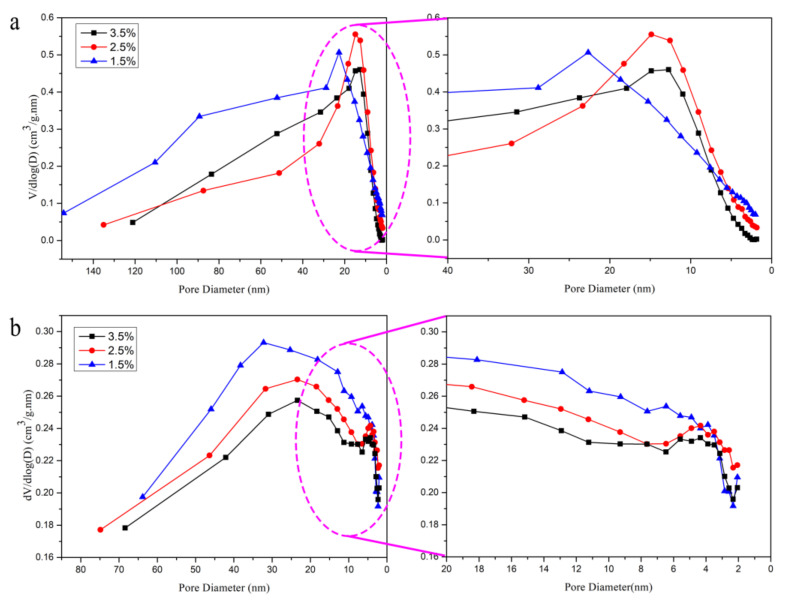
The pore size distributions of CNF aerogels via two kinds of drying methods: (**a**) supercritical CO_2_ drying method and (**b**) liquid nitrogen freeze-drying technology.

**Table 1 polymers-12-02583-t001:** The shrinkage, Brunauer-Emmett-Teller (BET), pore volume, and density of CNF aerogels via the supercritical CO_2_ drying method.

Samples	Shrinkage (%)	BET (m^2^/g)	Pore Volume (cm^3^/g)	Density (g/cm^3^)
1.5-SCD-CNF aerogel	5.56	296.23	0.7196	0.0201
2.5-SCD-CNF aerogel	4.12	324.36	0.6112	0.0263
3.5-SCD-CNF aerogel	4.35	330.88	0.5024	0.0362

**Table 2 polymers-12-02583-t002:** The shrinkage, BET, pore volume, and density of CNF aerogels by liquid nitrogen freeze-drying technology.

Samples	Shrinkage (%)	BET (m^2^/g)	Pore Volume (cm^3^/g)	Density (g/cm^3^)
1.5-FDN-CNF aerogel	12.63	106.25	0.8556	0.0363
2.5-FDN-CNF aerogel	8.56	162.92	0.6812	0.0502
3.5-FDN-CNF aerogel	7.78	175.85	0.6324	0.0542

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
