# Peer review of "Fabrication and Characterization of Cellulose Nanofiber Aerogels Prepared via Two Different Drying Techniques"

_polymers, 2020, doi:10.3390/polym12112583_

Round 1

Reviewer 1 Report

Polymers-991390

In this paper, the authors show the fabrication and characterization of cellulose nanofibers (CNF) prepared via two different techniques. The CNF was prepared using supercritical CO2 and liquid nitrogen, respectively. The two structures obtained are very similar. The microstructure, shrinkage, surface area, pore-volume, density, compression strength, and isothermal desorption were analyzed, and have compared these properties of the CNF obtained via the two methods.

This paper is the sufficiently quality for publication in Polymers but with minor revision

Comment

  • In XRD where appears the peaks of the cellulose II. Comment with major detail the polymorphism of the cellulose I and cellulose II
  • Complete the conclusions

Other comment

  • Line 75

CHANGE              persulfate (S2O82-)                          FOR        sulfate (SO42-)

Revise all manuscript

  • Line 96

Because the inorganic salt is CaCl2. Comment in the manuscript

  • Line 132

CHANGE              Ka                                                          FOR        Kα

  • 2

CHANGE              ν                                                            FOR        V

  • Line150-151

Delete: equation

  • Line 175

CHANGE              cm-1                                                     FOR        cm-1

Revise all manuscript

  • Revise the presentation of the Figure 3b
  • Line 197

Delete: “as shown in Figures 4(a-c)

  • Line 221

Delete: pores

  • Line 262

CHANGE              CO2                                                      FOR        CO2 (Revise all manuscript)

  • Line 339

CHANGE              N2                                                         FOR        N2 (Revise all manuscript)          

Author Response

In this paper, the authors show the fabrication and characterization of cellulose nanofibers (CNF) prepared via two different techniques. The CNF was prepared using supercritical CO2 and liquid nitrogen, respectively. The two structures obtained are very similar. The microstructure, shrinkage, surface area, pore-volume, density, compression strength, and isothermal desorption were analyzed, and have compared these properties of the CNF obtained via the two methods.

This paper is the sufficiently quality for publication in Polymers but with minor revision

  • Comment 1: In XRD where appears the peaks of the cellulose II. Comment with major detail the polymorphism of the cellulose I and cellulose II

Respond: Thank you for your valuable suggestions, which will guide us in the right direction for our next research. Cellulose I and cellulose II, named as native and regenerated cellulose, respectively, are the most common polymorphs of cellulose of interest. The hydrogen bonding in CII is more complex than CI and inter-plane hydrogen bonds along with inter-chain bonds are possible because of an anti-parallel chain model. Both CI and CII exhibit different features such as mechanical properties, hydrophilicity and oil/water interface of Pickering emulsion. CI has been widely used in several applications such as preparing hydrogel for wound dressing and reinforcing other polymers to improve mechanical properties. According to the reviewer’s suggestion, we further discussed this issue.
Comment 2: Line75"CHANGE         persulfate(S2O82-)                 FOR        sulfate (SO42-)".

Respond: We have corrected this error and checked similar errors throughout the manuscript.
Comment 3: Line 96" Because the inorganic salt is CaCl2. Comment in the manuscript ".

Respond: We have corrected this error and checked similar errors throughout the manuscript.
Comment 4: Line 96"CHANGE              Ka                      FOR        Kα".

Respond: We have corrected this error and checked similar errors throughout the manuscript.

Comment 5: 2"CHANGE              v                      FOR       V".

Respond: We have corrected this error and checked similar errors throughout the manuscript.

Comment 6: Line150-151, Delete: equation

Respond: We have revised this phase according to the reviewers’ comment.

Comment 7: Line175, "CHANGE             cm-1                      FOR       cm-1".

Respond: We have corrected this error and checked similar errors throughout the manuscript.

Comment 8: Revise the presentation of the Figure 3b.

Respond: We have revised this phase according to the reviewers’ comment.

Comment 9: Line197, Delete: as shown in Figures 4(a-c).

Respond: We have revised this phase according to the reviewers’ comment.

Comment 10: Line221, Delete: pores.

Respond: We have revised this phase according to the reviewers’ comment.

Comment 11: Line262, "CHANGE             CO2                      FOR       CO2".

Respond: We have corrected this error and checked similar errors throughout the manuscript.

Comment 12: Line339, "CHANGE             N2                      FOR       N2".

Respond: We have corrected this error and checked similar errors throughout the manuscript.

Reviewer 2 Report

The manuscript “Fabrication and Characterization of Cellulose Nanofiber Aerogels Prepared via Two Different Drying Techniques” deals with the production of biocompatible and biodegradable aerogels, comparing different techniques.

The work is well organized and numerous and relevant characterizations have been performed on the obtained cellulose-based aerogels. However, before publication, some revisions are required.

In particular:

-  Introduction. The state of art investigation can be enlarged, in order to better highlight the advantages of the supercritical CO2 drying. Several recent works can be found in the literature; for example, the work of Baldino et al., Natural aerogels production by supercritical gel drying, Chemical Engineering Transactions, 2015, 43, pp. 739-744; etc…

-  Results and discussion. The authors selected only a couple of operative conditions; i.e., 120 bar and 40 °C. As it is known from the literature, the main requirement to obtain a successful supercritical drying is the formation of a supercritical mixture (CO2+organic solvent), characterized by a near to zero surface tension. Operating in this way, the collapse of the delicate gel structure is avoided. However, it is not clear why these operative conditions were selected in this work and if they have already been optimized. Other couple of pressure and temperature can be tried in order to determine a possible effect on aerogel porosity and performance. Solvent residues analysis can be also performed to verify if the drying was complete.

-  An extensive comparison between freeze drying and supercritical drying, pointing out the advantages/disadvantages of these processes, can help the reader to understand the aim of this work.

Author Response

The manuscript “Fabrication and Characterization of Cellulose Nanofiber Aerogels Prepared via Two Different Drying Techniques” deals with the production of biocompatible and biodegradable aerogels, comparing different techniques.

The work is well organized and numerous and relevant characterizations have been performed on the obtained cellulose-based aerogels. However, before publication, some revisions are required.
Comment 1: Introduction. The state of art investigation can be enlarged, in order to better highlight the advantages of the supercritical CO2 drying. Several recent works can be found in the literature; for example, the work of Baldino et al., Natural aerogels production by supercritical gel drying, Chemical Engineering Transactions, 2015, 43, pp. 739-744; etc…

Respond: The reviewer is sincerely thanked for the comments and suggestions. We have downloaded the paper (Baldino et al, 2015, 43,739) and carefully read the typical references. We have added this reference and supplied the advantages of this paper in the Introduction section.

Comment 2: Results and discussion. The authors selected only a couple of operative conditions; i.e., 120 bar and 40 °C. As it is known from the literature, the main requirement to obtain a successful supercritical drying is the formation of a supercritical mixture (CO2+organic solvent), characterized by a near to zero surface tension. Operating in this way, the collapse of the delicate gel structure is avoided. However, it is not clear why these operative conditions were selected in this work and if they have already been optimized. Other couple of pressure and temperature can be tried in order to determine a possible effect on aerogel porosity and performance. Solvent residues analysis can be also performed to verify if the drying was complete.
Respond: Thank you for your valuable suggestions, which will guide us in the right direction for our next research. These are the conditions that have been optimized. The relevant conditions have been determined by the research group. Relevant results are as follows: Wang, et al. Fabrication and characterization of nano-cellulose aerogels via supercritical CO2 drying technology. Materials Letters 2016, 183, 179. 29.30.   Zhang, T. et al. Characterization of the nano-cellulose aerogel from mixing CNF and CNC with different ratio. Materials Letters 2018, 229, 103.

Comment 3: An extensive comparison between freeze drying and supercritical drying, pointing out the advantages/disadvantages of these processes, can help the reader to understand the aim of this work.
Respond: Thank you for your valuable suggestions, which will guide us in the right direction for our next research. We have inserted a wide comparison between freeze-drying and supercritical drying in the introduction. And point out the advantages/disadvantages of these processes.